# Synchrotron-Based Fourier-Transform Infrared Micro-Spectroscopy of Cerebrospinal Fluid from Amyotrophic Lateral Sclerosis Patients Reveals a Unique Biomolecular Profile

**DOI:** 10.3390/cells12111451

**Published:** 2023-05-23

**Authors:** Tanja Dučić, Jan Christoph Koch

**Affiliations:** 1CELLS−ALBA, Carrer de la Llum 2-26, Cerdanyola del Valles, 08290 Barcelona, Spain; 2Department of Neurology, University Medicine Göttingen, Robert-Koch-Str. 40, 37075 Göttingen, Germany

**Keywords:** Amyotrophic lateral sclerosis, biomarker, cerebrospinal fluid, Fourier-transform infrared spectroscopy

## Abstract

Amyotrophic lateral sclerosis (ALS) is a fatal neurodegenerative disease, with the most common adult-onset neurodegenerative disorder affecting motoneurons. Although disruptions in macromolecular conformation and homeostasis have been described in association with ALS, the underlying pathological mechanisms are still not completely understood, and unambiguous biomarkers are lacking. Fourier Transform Infrared Spectroscopy (FTIR) of cerebrospinal fluid (CSF) is appealing to extensive interest due to its potential to resolve biomolecular conformation and content, as this approach offers a non-invasive, label-free identification of specific biologically relevant molecules in a few microliters of CSF sample. Here, we analyzed the CSF of 33 ALS patients compared to 32 matched controls using FTIR spectroscopy and multivariate analysis and demonstrated major differences in the molecular contents. A significant change in the conformation and concentration of RNA is demonstrated. Moreover, significantly increased glutamate and carbohydrates are found in ALS. Moreover, key markers of lipid metabolism are strongly altered; specifically, we find a decrease in unsaturated lipids and an increase in peroxidation of lipids in ALS, whereas the total amount of lipids compared to proteins is reduced. Our study demonstrates that FTIR characterization of CSF could represent a powerful tool for ALS diagnosis and reveals central features of ALS pathophysiology.

## 1. Introduction

Amyotrophic lateral sclerosis (ALS) is a fatal neurodegenerative disease characterized by the degeneration of upper descending cortical neurons and lower motoneurons located in the brainstem and spinal cord [1]. This results in progressive muscle paresis not only in the legs and arms but also in the thoracal and bulbar regions. Histopathologically, aggregates containing the TAR DNA-binding protein (TDP)-43 can be found in neurons [2], and a spreading pattern of those aggregates over the central nervous system has been proposed [3]. The pathomechanisms underlying ALS are only poorly understood at the moment [4,5]. About 90% of ALS patients are sporadic, while the remaining 10% of cases are inherited forms called familial ALS. Mutations in more than 10 genes have been shown to cause familial ALS but can also be detected in up to 15% of sporadic patients, including SOD1 (superoxide dismutase 1) [6], TDP-43 [2], FUS (fused in sarcoma) [7], and C9orf72 (chromosome 9, open reading frame 72) [8]. Accumulating evidence indicates that different mechanisms are responsible for the death of motoneurons in ALS, such as the deposition of insoluble aggregated, phosphorylated and cleaved TDP-43 protein in the cytoplasm of motoneurons, impaired autophagy, disrupted axonal transport and dysregulated RNA metabolism [9]. Interestingly, patients often show hypermetabolism starting before the onset of motor symptoms, involving marked changes in lipid and glucose metabolism [10].

Up until now, no fully specific biological markers for ALS have been identified [11]. Diagnosis depends on recognizing a characteristic clinical constellation that includes both upper and lower motor neuron degeneration as well as progressive motor dysfunction after careful exclusion of other conditions. Neurofilaments (light (Nf-L) and phosphorylated heavy chain (pNfH)) are elevated in serum and cerebrospinal fluid (CSF) in ALS patients and have a 90% sensitivity and specificity for the diagnosis of ALS in clinically pre-selected patient cohorts [12]. However, they are also elevated in other neurodegenerative diseases, such as Parkinson’s syndrome and dementia. The level of neurofilaments in ALS corresponds with disease progression, meaning that high levels indicate a fast progression and low levels a rather slow progression [13,14,15]. Other biomarkers, such as p75^ECD^ [16] and chitotriosidase 1 [17], have been proposed but are not yet widely used in clinical practice. 

Therapy has long been limited to symptomatic treatments and the glutamate inhibitor riluzole, which has moderate effects on survival. Recently, new therapeutic approaches targeting oxidative stress (e.g., edaravone [18]), mitochondria and endoplasmatic reticulum (e.g., sodium phenylbutyrate and taurursodiol [19]), as well as antisense oligonucleotides targeting specific gene mutations (e.g., tofersen [20]), were shown to have some effect on disease progression, but so far no therapy has been able to stop or even cure ALS. Thus, there is a great need to better understand the pathomechanisms involved in ALS as well as to identify more reliable biomarkers for diagnosis and estimation of disease severity.

Evaluation of the CSF is part of the standard diagnostic workup for ALS, mainly to exclude other conditions. The CSF is particularly interesting since it is in direct contact with the affected neurons and thus contains specific fingerprints of the pathological process. This is already well established for Alzheimer’s disease [21] (elevated tau and decreased Amyloid-beta-42) and Parkinson’s disease [22] (decreased alpha-synuclein), for example. Moreover, in ALS, several interesting specific findings in the CSF have already been reported [23]. Interestingly, several studies have found cytotoxic effects of CSF from ALS patients [24,25], suggesting a role for the CSF itself in ALS pathogenesis.

Synchrotron-based Fourier-Transform Infrared Spectroscopy (SR-FTIR) is a non-destructive, well-established technique that enables the association of vibrational peaks of the IR absorption spectra with specific chemical groups. This allows the characterization of specific bio-macromolecules (nucleic acids, carbohydrates, proteins, lipids) in cells, tissues and body fluids [26]. Due to the synchrotron infrared source, which is up to 1000 times brighter than a conventional thermal source, a higher spatial resolution and spectral quality can be achieved compared to classic FTIR. Combined with multivariate analysis and principal component analysis (PCA), SR-FITR is a potent diagnostic tool to achieve a rapid spectroscopic fingerprint of a sample containing information on the composition and structure of all main biomolecules and to compare different spectral data sets for discriminant features. A big advantage of FTIR is that invasive preparation protocols, denaturation of the essential bio-macromolecules, and other chemical changes caused by usual drying or fixation protocols are avoided. Previously, we provided SR-FTIR analysis of single astrocytes in a rat SOD1-model for ALS [27], as well as recently analyzing the co-localization of lipids with metals in the same model [28]. In another work, misfolded TDP-43 could be detected in human CSF samples with reasonably high specificity and sensitivity using an immune-infrared sensor [29]. FTIR of human tear fluid showed a highly specific spectral pattern that discriminated ALS patients from healthy controls [30]. 

Here, we present a comprehensive SR-FITR analysis of CSF from ALS patients in comparison to age-matched controls. The goal of the study was to determine the specific biomolecular fingerprint of ALS in CSF in order to obtain more insights into the pathophysiology and explore the method as a putative biomarker and diagnostic tool.

## 2. Materials and Methods

### 2.1. CSF Sampling

CSF was collected at the clinic for neurology at the University of Medicine Göttingen (UMG), Germany. Permission from the local ethics committee was given prior to the initiation of the study (Ethics Committee of the UMG, No. 13/11/12). Written consent was obtained from all patients. The study conforms to the Code of Ethics of the World Medical Association (Declaration of Helsinki). 

Lumbar puncture was performed according to standard procedures, and 10 mL of CSF per patient were collected in polypropylene tubes. One 2 mL CSF sample together with a corresponding serum sample was evaluated in the local CSF lab at UMG for analysis of routine CSF parameters, including leukocyte and erythrocyte count, total protein, albumin, lactate, concentrations and CSF/serum quotients of IgG, IgM and IgA, as well as oligoclonal bands. Samples with a leukocyte count > 6/µL or a red blood count > 20/µL in routine testing were excluded from the study. The remaining 8 mL of all samples were immediately centrifuged (within 30 min after collection) at 2000× *g* at 4 °C for 20 min and frozen at −80 °C until further analysis. Frozen samples were collectively transported on dry ice from Göttingen to the ALBA synchrotron facility in Barcelona, Spain, where SR-FTIR was performed at the MIRAS beamline (project AV 2019093794). A volume of 3 µL of the CSF was placed on 0.5 mm thick CaF_2_ glass prior to the measurements.

### 2.2. Participants

CSF from 33 ALS patients and 32 age- and gender-matched controls were included in the study. For ALS patients, the diagnosis was at least “clinically probable laboratory supported” according to the revised El Escorial criteria [31]. They were arbitrarily recruited from the available patient pool of the out- and in-patient clinics. All patients underwent a thorough clinical work-up, including electrophysiological testing (electromyography, neurography, evoked potentials), MRI of the brain and spinal cord, and blood laboratory testing. Several follow-up visits confirmed the diagnosis and disease progression. Patients were classified as “bulbar” or “spinal” depending on the region that was affected first. Clinical symptoms were rated using the revised ALS Functional Rating Scale (ALSFRS-R) [32]. The ALSFRS-R Slope was calculated by dividing the number of ALSFRS-R points lost at the time point of lumbar puncture by the disease duration since the first symptom onset (paresis, muscle atrophy, dysarthria, or dysphagia) in months. In all but one patient, the neurofilament phosphorylated heavy chain (pNfH) in the CSF was evaluated.

As controls, CSF was drawn from 32 age- and gender-matched subjects who presented with neurological disorders other than ALS. They received a lumbar puncture mainly for other conditions (e.g., headache, dizziness, polyneuropathy, syncope) as exclusion diagnostics. A few patients with typical ALS mimics, such as spinal muscular atrophy (1), cerebellar ataxia (1), polyneuropathy (5), multiple sclerosis (2) and spinal canal stenosis (1), were also included in the control cohort since they constitute the usual patient group that qualifies for differential diagnosis of motoneuron disease. It was made sure that the basic CSF parameters, including leucocyte cell count, erythrocytes, protein, lactate and intrathecal Ig-synthesis, were in the normal range for all control patients, therefore excluding severe inflammatory conditions of the CSF.

Patient characteristics, including basic CSF parameters, are shown in detail in Appendix A. Taken together, the mean age of the ALS patients was 63 ± 9 years, as compared to 61 ± 15 years for the control patients. Of the 33 ALS patients, 21 were male (63.6%) and 12 were female, while of the 32 control patients 19 were male (59.4%) and 13 were female. This corresponds very well with the previously published epidemiological data of ALS [33] in a broader population where age and gender are very similar and distributed to our study population. Nine of the 33 ALS patients had a bulbar onset (27%), which is more than expected in the broader population (around 15% of all ALS patients). At the time of the lumbar puncture for the ALS patients, the mean disease duration was 16 ± 15 months (range 3–72 months), the mean ALSFRS-R score was 40 ± 6 points (out of 48 points) (range 18–47), and the mean ALSFRS-R Slope was 1 ± 0.6 points per month. (range 0.1–2.6) With regards to pNfH in the CSF, the mean value of the ALS patients was 3220 ± 2466 pg/mL, the median was 2319 pg/mL and the range was 244–10,518 pg/mL (the norm was 62–553 pg/mL) [12]. Three of the included ALS patients (10%) showed a repeat expansion mutation in the C9orf72 gene; no other genetic mutations were found in the study population. 

Basic CSF parameters were unremarkable for both groups. ALS patients had a mean leukocyte count of 1 ± 1 cells/µL (range: 0–6), a mean total protein concentration of 430 ± 153 mg/L (range: 243–949), a mean albumin quotient of 7 ± 3 (range: 3.6–17.9) and a mean lactate of 2 ± 0.4 mmol/L (range: 1.4–3.6). Control patients had a mean leukocyte count of 1 ± 1.4 cells/µL (range: 0–5), a mean total protein concentration of 485 ± 209 mg/L (range: 176–1142), a mean albumin quotient of 8 ± 4.4 (range: 2–23) and a mean lactate of 2 ± 0.2 mmol/L (range: 1–2).

### 2.3. Synchrotron-Based FTIR Analysis

The SR-FTIR measurements were conducted at the MIRAS beamline at the ALBA synchrotron, Barcelona, as the synchrotron light was used as the infrared light source, coupling with the 3000 Hyperion microscope coupled to a Vertex 70v spectrometer and a liquid nitrogen-cooled mercury cadmium telluride (MCT) detector. The spectroscopic data were collected in transmission mode using the 36X Schwarzschild objective and condenser and an aperture size of 10 µm × 10 µm. Overall, 4–6 repetitions per measurement per sample were made, and average spectra are presented. The data were collected in the 4000–800 cm^−1^ mid-infrared range at a spectral resolution of 4 cm^−1^ with 256 co-added scans per spectrum. The OPUS 8.2 (Bruker, Ettlingen, Germany) software package was used for data acquisition.

Atmospheric subtraction, rubber band baseline correction and vector normalization were performed for each single spectrum. For easier interpretation, three different areas were analyzed: the 3050–2800 cm^−1^ lipids area, 1800–1480 cm^−1^ proteins and esters region, and 1480–900 cm^−1^ nucleic acids and carbohydrates region. The positions of the peaks in the spectral analysis were determined using the second derivative method by the Quasar software (Bioinformatics Laboratory of the University of Ljubljana, Slovenia [34], Version 1.7.0 with the spectroscopy package [35]), and it was also used to perform the principal component analysis (PCA) and peaks deconvolution. 

The ratios of bands characteristic for lipid oxidation were calculated for asymmetric CH_2_ (2900–2945 cm^−1^) and CH_3_ groups (2945–2990 cm^−1^). The integrated areas for total proteins (1500–1700 cm^−1^), lipids (2800–3050 cm^−1^) and asymmetric CH_3_ were calculated by integration of assigned areas. The deconvolution of the amide I and II and ester area (1800 to 1530 cm^−1^) of SR-FTIR spectra was performed by using least-squares iterative curve fitting to Gaussian line shapes in Quasar software. Spectral analysis where the second derivative was performed (21 smoothing points, 3rd polynomial order and vector normalization), e.g., for the fingerprint region, was only vector-normalized, as this normalization technique does not require a reference peak [26]. 

Statistical tests were performed using the Student’s *t*-test in the OriginPro 2019 software (Northampton, MA, USA). Differences were considered significant with *p* < 0.05.

## 3. Results

### 3.1. Comparison of Complete Spectra

We performed SR-FTIR with the CSF of 33 ALS patients and 32 controls. After the deposition of CSF samples on the IR transparent CaF_2_ support and excess water evaporation, the CSF fluid displayed a typical fern-like morphology (Appendix A), surrounded by an amorphous drop border. The complete FTIR averaged spectra of the CSF samples of ALS and the control group in the region 4000–900 cm^−1^ are depicted in Figure 1, with the major bands annotated.

In order to analyze in detail all relevant biomolecules, we split the spectra into two major parts: the fingerprint area (900–1800 cm^−1^) and the lipid part (2800–3000 cm^−1^). Further, the fingerprint area is divided into two smaller groups: the proteins and ester group (1480–1800 cm^−1^) and the nucleic acids and carbohydrates group (900–1480 cm^−1^) (Figure 1).

In the so-called fingerprint area, we describe the main bands associated with nucleic acids, i.e., phosphate molecule vibrations (DNA ~1030 and 1220 cm^−1^) [36] and RNA (~1120 cm^−1^) [37] and also the COH deformation band and C–O stretching bands of carbohydrates (~1056, 1078 and 1150 cm^−1^, respectively) [36]. The main protein bands Amide I and Amide II ~1660 and 1540 cm^−1^ are analyzed in the second part of the results, and the exact band position is determined by the backbone conformation and the hydrogen bonding pattern, thus resolving the secondary structure of proteins. Finally, lipid metabolism is presented in the third part as associated with C–H band vibrations in the range of 2800–3000 and 1455 and 1340 cm^−1^, as well as the carboxyl and carbonyl groups (1400 and 1740 cm^−1^, respectively) [36].

### 3.2. “Fingerprint” Area of the CSF Spectra

The spectral range between 900 and 1800 cm^−1^ is very complex due to the coinciding different biomolecules, such as carbohydrates and phosphates, which are constituent parts of the nucleic acids, i.e., RNA and DNA, but could also reflect biochemical processes such as phosphorylation and glycosylation. In this so-called “fingerprint area”, several bands were significantly different in the CSF of ALS patients in comparison to control samples. To make precise peak positions, the second derivative was performed on the assigned area, which includes proteins, esters, nucleic acids and carbohydrates as shown in Figure 2A. Major differences were observed concerning the bands assigned in Figure 2B. The ALS samples showed more pronounced peaks at ~1665, 1587, 1445, 1254, 1120, 1030 and 920 cm^−1^. These bands are associated with Amide I (a characteristic H–N–C=O bond within proteins), the amino acid glutamic acid or glutamate (Glu, according to the NIST standard and [38]), the DNA band, the RNA band, carbohydrates and the Z-conformation of DNA, respectively. On the other hand, the control samples had more pronounced carbonyl groups (band at ~1740 cm^−1^), β-sheet structure (1635 cm^−1^) and carboxyl groups at 1400 cm^−1^. The band at ~1110 cm^−1^ is connected to different carbohydrate moieties such as ribose and deoxyribose [39] and/or phosphate [30]. The band of COH deformation at 1052 cm^−1^, in sugars, and the band at 970 cm^−1^, assigned to phosphorylated proteins, were also higher in control CSF samples (Figure 2B). 

The PCA analysis (Figure 2C,D) showed the PCA score segregation mostly over the PC1. The first principal component pointed out the differences mainly in the secondary protein structure, with the minima at 1653 and 1545 cm^−1^ (both assigned to α-helix) abundant in the control samples and the maxima (1695 and 1580 cm^−1^, β-antiparallel structure and amino acid Glu, respectively), present mostly in ALS samples. Besides, the control samples showed more pronounced bands at 1101 and 1063 cm^−1,^ corresponding to the nucleic acids and the –CO–O–C stretching band in cholesterol esters [36], respectively. The peak at ~1101 cm^−1^ pointed out differences in the process of DNA methylation [40]. On the contrary, the maxima at 1082 cm^−1^ associated with phosphate symmetric stretching vibrations and at 1124 cm^−1^ corresponding to RNA, as well as the band at ~920 cm^−1^, associated with the Z-form of DNA [41], are linked with the ALS samples (Figure 2D). 

The PC2 describes 20% of the total variability in the “fingerprint” area and points towards differences mainly at 1652, 1545 and 960 cm^−1^. The band at ∼960 cm^−1^ is assigned to *γ*_as_-N(CH_3_)_3_− the vibration of choline, and it was more prominent in the control spectrum in comparison to ALS samples (Figure 2D). Choline and its metabolites are, besides their role in maintaining the structural integrity of the cell membrane, part of acetylcholine, which is the essential neurotransmitter at the neuromuscular junction. Since the neuromuscular junctions are among the first cellular sites to be affected by degeneration in ALS, one could assume reduced acetylcholine levels in ALS patients. Our data indeed showed a decrease of choline in ALS CSF, corresponding to a decrease of the area under the peak from 0.49 ± 0.4 to 0.38 ± 0.3; however, this change was not statistically significant.

### 3.3. Nucleic Acids, Carbohydrates and Phosphates

Additionally, to go more in detail and evaluate the bands of the region of the nucleic acids and carbohydrates, we performed the deconvolution of the area from 900 to 1180 cm^−1^, where the main peaks at ~930, 980, 1035, 1080, 1105, 1120 and 1160 cm^−1^ were evaluated. The wave number ~930 cm^−1^ corresponds to bands related to the Z-form, a conformationally changed form of DNA [41]. The wave number ~984 cm^−1^ could be assigned to uracil ring motions of RNA [42] or phosphorylated proteins [36]. The C–OH stretching vibration at ~1030 cm^−1^ corresponds to nucleic acids [36], while the band at ~1080 cm^−1^ is more complex and could be assigned to PO_2_^−^ symmetric stretching vibrations (~1085 cm^−1^) and C–C movement in carbohydrates (peak at 1078 cm^−1^) [36]. The C–O stretching vibration of the ribose ring of RNA corresponds to the peak at ~1120 cm^−1^ [37] and the C–O of proteins and carbohydrates (1160 cm^−1^) [42]. 

The deconvoluted peaks revealed a significant difference at the bands 984 cm^−1^ (Figure 3B) and 1078 cm^−1^ (Figure 3C). Interestingly, the band at 1078 cm^−1^ assigned to carbohydrates [36,39] was significantly higher in ALS samples, indicating a difference in carbohydrate content in ALS.

RNA-specific infrared absorptions at 1125 (±7 cm^−1^) and 984 cm^−1^ [43] are particular peaks considered diagnostic of the contribution of RNA in the nucleic acid backbone vibrations [43,44,45]. Both these peaks showed significant changes for the calculated area at 984 cm^−1^ (Figure 3B) and the peak position at 1120 cm^−1^ (Figure 3D), suggesting changes in RNA metabolism and a significant decrease of the whole RNA amount in ALS samples. A significant difference was observed in the peak position (shifted to a lower wavenumber) in ALS samples in comparison to control samples, pointing out changes in the structure. Consequently, the ratio of DNA vs. RNA was higher in ALS samples, where RNA quantity was calculated as the area under the peak at 1120 cm^−1^ and DNA area under the band at 1220 cm^−1^, almost reaching significance (*p* = 0.059) (Figure 3E).

### 3.4. Proteins

The FTIR spectra from Amide I and Amide II, as well as the ester groups (~1740 cm^−1^), are presented in Figure 4A. The spectral profiles of Amide I and II differed significantly. As shown in Figure 4A, the spectra of ALS samples changed shape at ~1580–1590 cm^−1^ in comparison to control samples. This peak shoulder corresponds to free amino acid side chains, specifically glutamate and/or aspartate [38]. The PCA analysis showed main differences over PC1 (Figure 4B), and bands at ~1705 and 1740 cm^−1^ were more pronounced in the control, while 1585 and 1657 cm^−1^ were more prominent in ALS samples (Figure 4C).

To the extent of protein analysis, the second derivative of the spectral range 1480–1800 cm^−1^ was calculated, and it showed changes in the specific secondary structure of proteins (Figure 5A). The most prominent bands in the CSF samples appear at 1740, ~1671, ~1662, ~1630, 1585, 1547 cm^−1^ corresponding to carbonyl groups, α-helix, turn and loops, β-sheet structures, and free amino acid side-chains [46,47], respectively. As assigned in Figure 5A, these bands differ between the two groups. In the Amide I area, the shape of the main peak around 1670 cm^−1^ was clearly different in ALS CSF samples. To find differences in sub-peak content, the protein region was analyzed in detail after the deconvolution of that area (Figure 5B). In that area, the second derivative showed hidden peaks, which were different from the control ones. After all spectra deconvolution, significant differences were found in the Glu amino acid side chains, as presented in panel 5C. The control samples showed a small peak at 1662 cm^−1^, which was lacking in the ALS samples. 

### 3.5. Lipids

In the region of lipids, the main differences concern the bands at ~2870 and ~2960 cm^−1^ assigned to *ν*_s_CH_3_ and *ν*_as_CH_3_ [36] (marked with arrows in Figure 6A). The PCA showed the main segregation over the PC1 (Figure 6B), and the loading plots confirmed the wavelengths of 2854 cm^−1^ assigned to *ν*_s_CH_3_ and 2957 cm^−1^ assigned to *ν*_as_CH_3_.

The calculated ratio of the two asymmetric bands CH_2_ and CH_3_ and important parameters for lipid peroxidation or oxidative stress estimation are presented in Figure 6D. This ratio was significantly higher in the ALS samples, which is a direct consequence of the lower asymmetric CH_3_ band in those samples (Figure 6E). Furthermore, the amount of unsaturated C=C bands significantly dropped in the ALS samples in comparison to the control (Figure 6F).

Besides, the calculated ratio of the integrated area of total lipids and proteins, which represent their relative concentrations, is shown in Figure 6G, and the ALS samples exhibited a significantly lower ratio.

### 3.6. Subgroup Comparisons

We speculated that age or sex might have a significant effect on the FITR spectra; however, this was not the case in our cohort, as shown in Appendix A.

High neurofilament levels in the CSF are correlated with a more aggressive course of ALS [13,14,15]. Thus, we analyzed changes comparing CSF samples from patients with relatively high (pNfH > 5000 pg/mL) versus low neurofilament (pNfH < 2000 pg/mL) levels in the CSF and control samples. Indeed, the spectra of patients with high pNfH were clearly more different from the control than the low pNfH samples, indicating more pronounced changes on the molecular level (Appendix A). Significant differences were found particularly for carbohydrates at 1078 cm^−1^, and also for Glu/Asp at 1587 cm^−1^, and a significantly lower concentration was found in control samples for both macromolecules, as presented in Appendix A, respectively. These findings fit well with the assumption that higher neurofilament levels indicate a more intensified disease pathology.

No significant differences for specific bands could be detected for fast (ALSFRS-R-Slope > one point per month (incl. 1)) versus slow (ALSFRS-R Slope < half points per month) clinical progression as assessed by the ALSFRS-R (Appendix A). However, there were some very interesting trends. Similarly, there were no significant differences when comparing the early (<11 months after symptom onset) and later (>11 months after symptom onset) stages of the disease (Appendix A). This finding suggests that the molecular changes found in this study represent a specific fingerprint of ALS pathophysiology independent of the disease state.

Interestingly, we found significant differences between patients with the spinal vs. bulbar subtype of ALS, particularly regarding the protein structure (Appendix A). There was a significant difference in the α-helix band (~1664 cm^−1^), which was significantly higher in bulbar patients, while the β-turn and loops band (~1680 cm^−1^) was significantly more pronounced in samples from spinal ALS patients.

## 4. Discussion

In this study, sixty-five CSF samples collected from ALS and control subjects were compared in order to analyze biomolecular changes using SR-FTIR spectroscopy. The analysis showed significant differences in the nucleic acids, carbohydrates, proteins and lipid metabolism in ALS samples in comparison with the control samples.

### 4.1. Nucleic Acids and Carbohydrates

Regarding the nucleic acid status, we found significant changes regarding the RNA-specific IR spectral bands at 1120 cm^−1^ and 984 cm^−1^, which are particular peaks considered diagnostic of the contribution of RNA in the nucleic acid backbone vibrations [43,44,45]. Our findings imply a decrease in the total amount of RNA in ALS CSF as well as specific changes in RNA metabolism. The position of the RNA peak was significantly shifted in ALS samples, indicating a three-dimensional conformation transition of RNA [48], which might be functionally relevant as RNA must fold into an accurate conformation for its proper function. Thus, the infrared spectral signature of the RNA is important to evaluate, as the shift in the RNA maximum absorption reflects the structural and conformational change of each biomolecule.

In recent years, it has become evident that RNA dysregulation plays a key role in ALS pathogenesis. The main ALS genes, TARDBP, FUS and C9orf72, are all strongly involved in RNA metabolism processes such as mRNA transcription, alternative splicing, RNA transport, mRNA stabilization and miRNA biogenesis [49,50]. Aberrant expression, dysfunction and particularly the aggregation of a group of RNA-binding proteins, including TDP-43, FUS and RBM45, are associated with several neurological disorders. These three disease-linked RNA-binding proteins all contain at least one RNA recognition motif [51].

Accordingly, the ratio of DNA vs. RNA relative concentration was higher in ALS samples. A very similar pattern was detected for phosphate assigned to nucleic acids earlier in astrocytes of the ALS SOD1-mouse model [27,28,52] and also recently in the tear fluid of ALS patients [30]. Very analogous results were found in the CSF of patients with multiple sclerosis; also, it was reported that the Z-form of DNA was found in samples from multiple sclerosis patients [53], thus suggesting a comprehensive role of the Z-form of DNA in different neurodegenerative diseases.

Regarding the carbohydrates, the band at 1078 cm^−1^ was significantly higher in ALS samples, implying differences in carbohydrate metabolism. A previous similar finding in tear fluid was interpreted as a higher degree of glycosylation (and phosphorylation) of tear lipids and proteins in ALS [30]. Our finding fits very well with the known dysregulation of glucose metabolism in ALS [54]. Several studies using FDG-PET imaging of the central nervous system (CNS) in ALS patients have demonstrated a local glucose neuronal hypometabolism in the motor cortex attributed to the degeneration of the motoneurons, while glucose hypermetabolism was found in many other regions of the CNS, including the midbrain, cerebellum and spinal cord, which has been ascribed to activation of microglia and astrocytes [55]. Animal studies have revealed a reduction in glycolytic enzyme activity and a reduced amount of glycolytic metabolites in neurons in ALS model systems. It is thus possible that the increased concentration of carbohydrates found in the ALS CSF samples reflects this impairment of glucose metabolism in ALS.

### 4.2. Proteins

Interestingly, in the protein area, the band at ~1585 cm^−1^ was significantly higher in the CSF samples from ALS patients than in control CSF, pointing to an enrichment in the amino acids glutamate or aspartate [38,56,57].

Inconsistent results on glutamate levels in the CSF of ALS patients have been published. While a small study in 17 ALS patients did not detect higher values of glutamate or other major amino acids in the CSF [58], a larger study in 377 ALS patients found significantly increased glutamate levels in the CSF in more than 40% of the examined ALS patients [59]. Furthermore, it was reported that in severely progressing ALS cases, serum glutamate and aspartate levels were increased [60]. Since glutamate was one of the most significant findings in our study, one can assume that the high sensitivity of FTIR enables the detection of glutamate changes in an even larger group of patients and thus underlines the importance of glutamate changes in ALS pathophysiology.

L-glutamic acid, L-glutamine and L-alanine are the most abundant amino acids in the CSF (50–55% of total amino acids) [61]. Various studies have focused on the study of amino acid neurotransmitters in the CSF, and glutamate has been considered the best candidate to follow ALS disease progression [62]. Glutamate is known to be the major neurotransmitter of the corticospinal areas and certain spinal cord interneurons [13], but it is also thought to be a potentially neuro-excitotoxic compound. This is underlined by the fact that the well-established therapy riluzole primarily works via glutamate antagonism. A systemic defect in glutamate metabolism may underlie at least certain forms of ALS [57,63]. Changes in glutamate concentration in body fluids may be caused by abnormalities in the transport or activity of key enzymes in glutamate metabolism, which were shown to be at least partly induced by other neurotoxic agents detectable in the body fluids of ALS patients [64]. As amino acids in the CSF are derived from brain tissue and as the enzyme glutamic oxaloacetic transaminase (GOT) catalyzes the conversion of L-aspartate to L-glutamate, abnormal biosynthesis or degradation of this enzyme could influence glutamate concentration [60]. Accordingly, it was shown before in other neurodegenerative diseases, e.g., Alzheimer’s disease, that CSF L-glutamine levels were higher compared to control CSF as a consequence of altered GOT enzyme activity [61,65].

### 4.3. Lipids

FTIR data of the lipid area showed a significantly decreased concentration of asymmetric vibration of the CH_3_ band in ALS in comparison to the control (Figure 6). Consequently, the ratio of the asymmetric bands CH_2_ and CH_3_ was increased in the ALS samples, indicating altered lipid metabolism, increased formation of lipid vesicles [27], and/or enhanced lipid peroxidation. Moreover, the amount of unsaturated C=C bands was significantly decreased in the ALS samples in comparison to the control.

These results could be a consequence of the high rate of lipid metabolism in ALS, leading to high cell lipidic (or membrane) turnover [66,67,68,69]. It is well established that lipid metabolism in general is significantly increased in ALS already early in the disease, along with a cellular switch from glucose oxidation (which is decreased as described above) to fatty acid β-oxidation. This enhanced lipid metabolism leads to increased peroxidation and oxidative stress [70], as witnessed by our results. Several studies have also shown a decrease in unsaturated lipids in ALS, probably as a consequence of these changes in lipid metabolism. Interestingly, it was even demonstrated that neurotoxic reactive astrocytes secrete saturated fatty acids that further contribute to cell death [71]. In line with that, we previously reported an increase in lipids and lipid vesicles in astrocytes of an ALS rat model [27]. Additionally, in the SOD1^G93A^ mouse model, astrocytes exhibited increased lipid peroxidation and large lipid rearrangements, including the formation of membranous vesicles as well as lipid modifications [28]. In humans, a lower lipid-to-protein ratio was observed in ALS samples, similar to changes in lipid–protein balance in the CSF at different stages of Alzheimer’s disease [72]. Thus, the significant changes in lipid composition found in the FITR spectra of ALS CSF samples reflect truly severely altered lipid metabolism as a core feature of ALS pathophysiology.

## 5. Conclusions

FTIR spectroscopy is progressively used in clinics because it provides a low-cost, fast and label-free macromolecular profile of different biological samples. Besides, as a non-invasive diagnostic tool, it could be very useful for the differential diagnosis of neurodegenerative disorders. In this study, we applied FTIR microspectroscopy, coupled with multivariate analysis, for the characterization of CSF from ALS patients and compared them to the control group in order to gain more insights into the pathophysiology and establish a putative new biomarker.

We found several specific changes in the FTIR spectra of ALS CSF vs. control that indeed underline central aspects of ALS pathophysiology. The total amount of RNA was reduced in ALS CSF, in particular when compared to DNA. Moreover, the conformation of RNA is altered in ALS. The relative concentration of carbohydrates was significantly increased in ALS CSF as a possible consequence of decreased glucose metabolism or enhanced glycosylation. With regards to proteins, an increase in glutamine amino acid levels was found in ALS, suggesting a malfunctioning of the metabolism of this amino acid and highlighting the putative neuro-excitotoxic role of glutamate in ALS. Substantial alterations were also observed in lipid metabolism, specifically a significant decrease of unsaturated lipids in ALS and an increased CH_2_/CH_3_ ratio indicative of enhanced peroxidation and oxidative stress.

Our study demonstrates that FTIR characterization of CSF could represent a powerful tool for differential diagnosis. The correlation of spectroscopic data with molecular markers, such as those derived from multiomics approaches, and the possibility to discriminate among ALS-imitating diseases will represent an important further step for their translation into the clinic.

Our results are expected to promote a better understanding of the etiology of ALS and could lead to the novel characterization of biomarkers of ALS.

## Figures and Tables

**Figure 1 cells-12-01451-f001:**
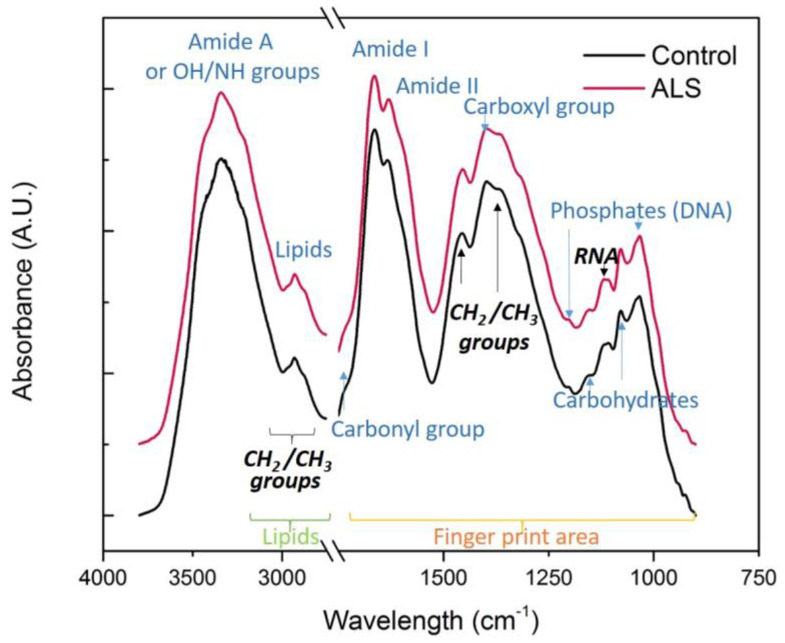
The complete FTIR averaged spectra of the region 4000–900 cm^−1^ of control CSF (black) and ALS samples (red). All spectra were baseline corrected and normalized, N = 32 and 33, respectively. The major bands are assigned.

**Figure 2 cells-12-01451-f002:**
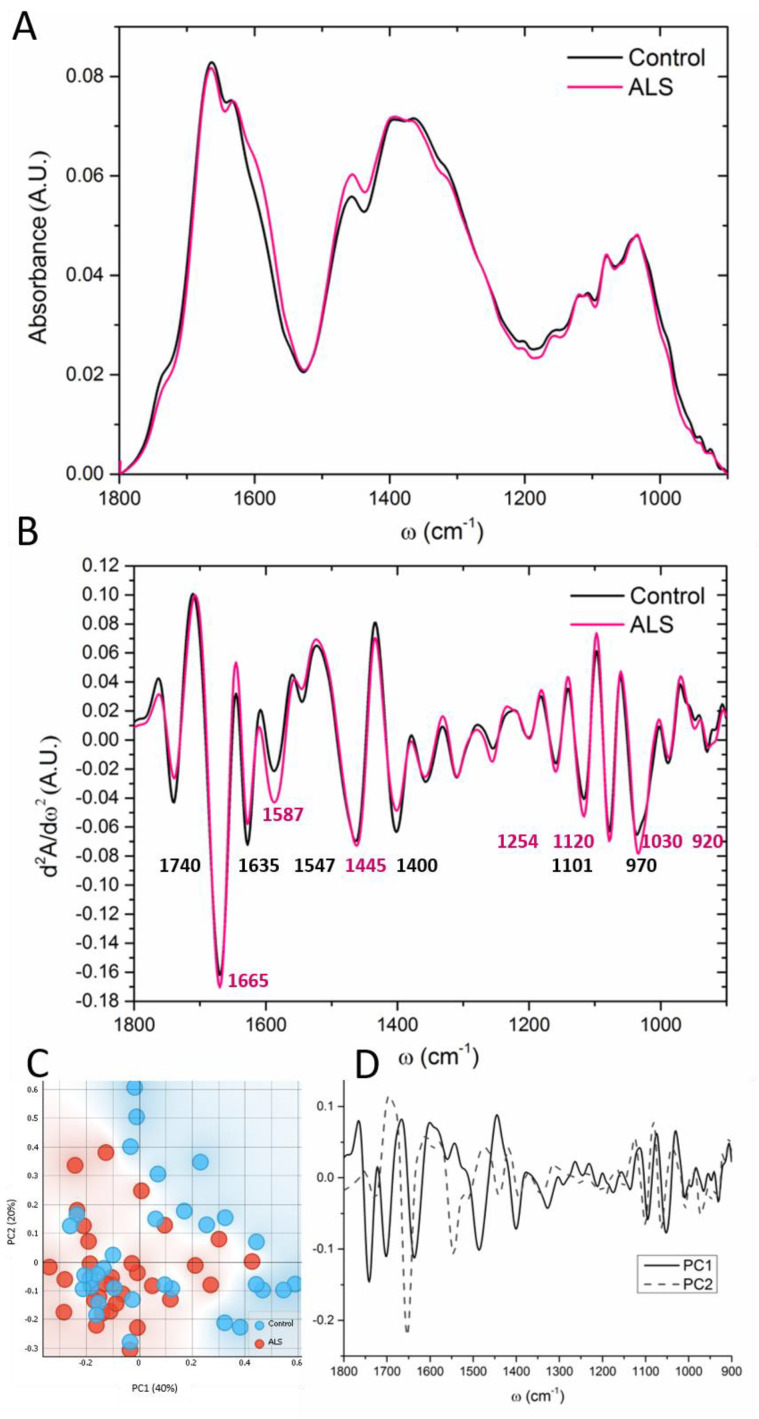
The averaged FTIR spectra (**A**) and second derivate of the finger print region 1800–900 cm^−1^ (**B**) of the FTIR averaged spectra of ALS (red) and control (black) CSF samples (N = 33, 32, respectively). The main peaks are denoted. (**C**) A scatter plot represents the PCA analysis of the first and second principal components; the graph in (**D**) shows the contribution of individual absorbance to the PCAs (loading values) of the first two principal components, PC1 and PC2.

**Figure 3 cells-12-01451-f003:**
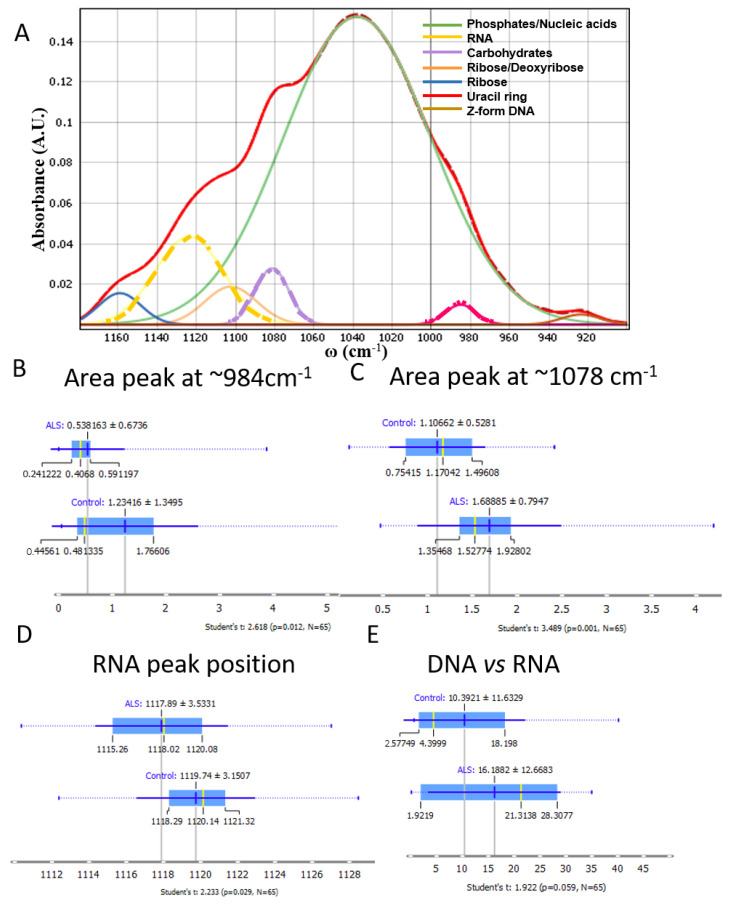
(**A**) An example of FTIR deconvoluted spectra of the CSF area of nucleic acids and RNA fit of the region 1180–900 cm^−1^. (**B**) Integrated area under the peak at ~984 cm^−1^ (red dash line). (**C**) Integrated area under the peak at ~1078 cm^−1^ (violet dash line); (**D**) the RNA peak position was calculated; (**E**) the ratio of the integrating area of the RNA peak band at 1120 cm^−1^ and the DNA (band at 1220 cm^−1^). (ALS N = 33, Control N = 32). *p* < 0.05 represents a significant difference.

**Figure 4 cells-12-01451-f004:**
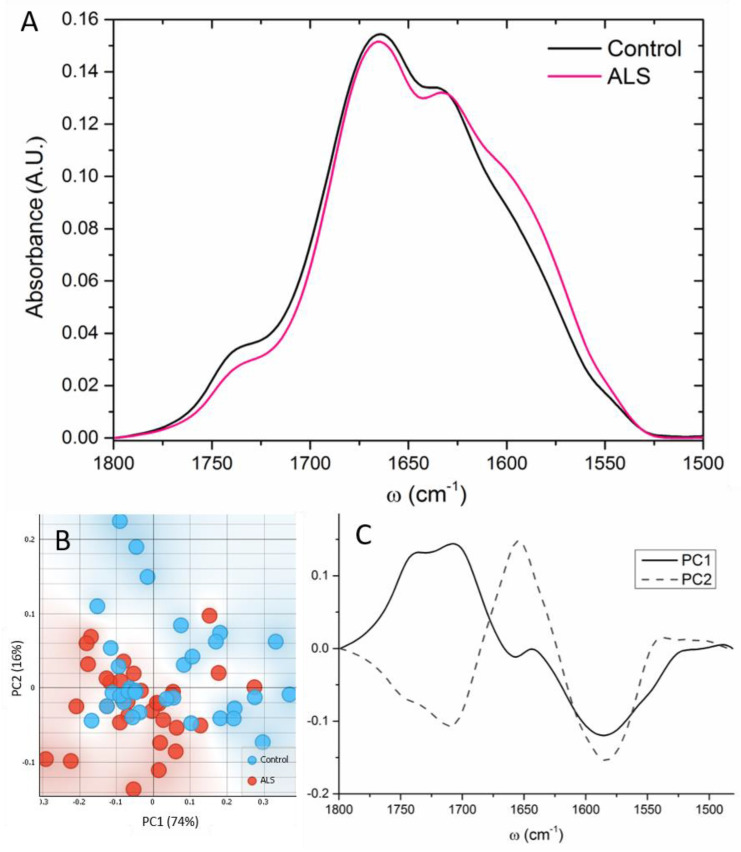
The complete FTIR averaged spectra of Amide I and Amide II and ester groups in the region 1480–1700 cm^−1^ (**A**) of ALS (red) and control (black) CSF samples (N = 33, 32, respectively). (**B**) Scatter plot represents the PCA analysis, where the first and second principal components are plotted. (**C**) The contribution of individual absorbance to the PCAs (loading values) of the first two principal components, PC1 and PC2.

**Figure 5 cells-12-01451-f005:**
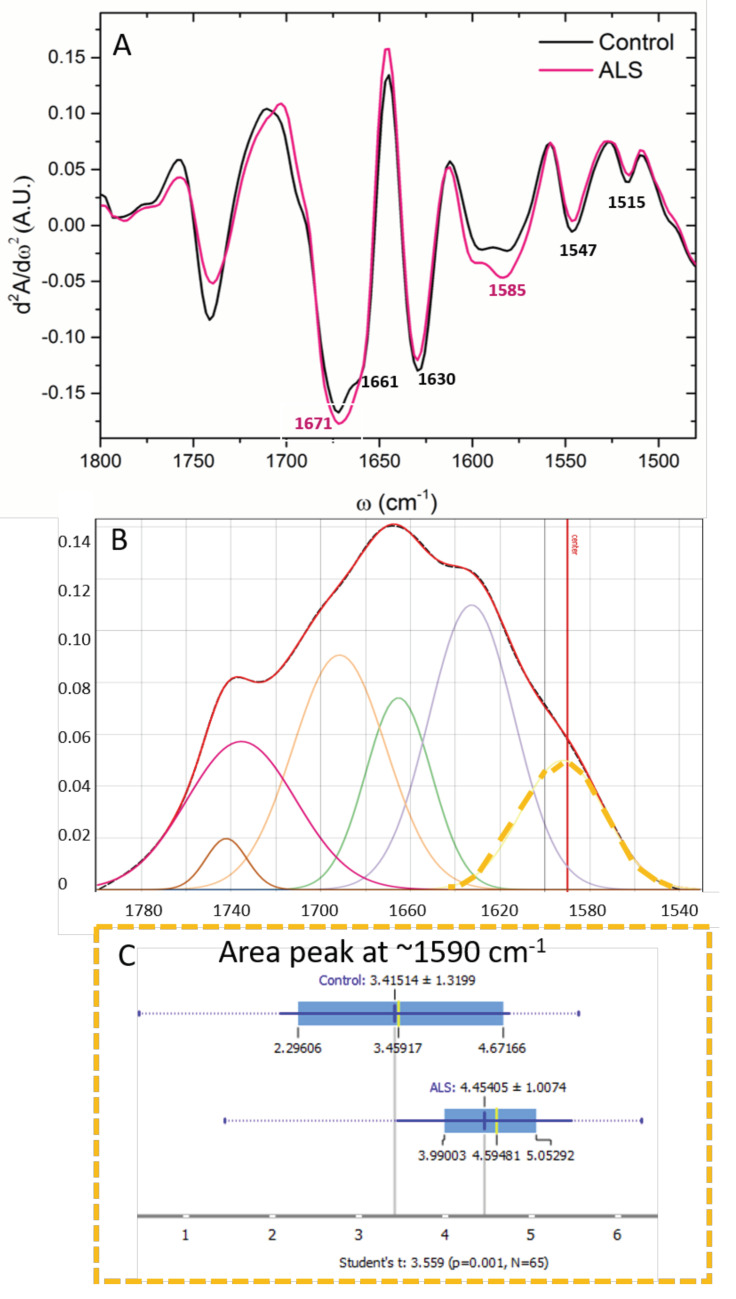
The second derivate of the spectral region 1480–1700 cm^−1^ of the FTIR averaged spectra of ALS (red) and control (black) CSF samples (**A**). An example of FTIR deconvoluted spectra of the CSF area of proteins fits the region 1520–1780 cm^−1^ (**B**). Secondary protein structure assignments of Amides I and II and ester groups are shown in different colors: bands corresponding to peaks at ~1740 cm^−1^ carbonyl group (brown line), ~1725 cm^−1^ carboxyl group (red), ~1690 cm^−1^ anti-parallel β-sheets (orange), ~1665 cm^−1^ (green) turns and loops and α-helix structures, ~1635 cm^−1^ parallel β-sheets (blue), and ~1585 cm^−1^ free amino chains (yellow). Integrated area under the peak at 1585 (yellow dash line). (ALS N = 33, Control N = 32) (**C**). *p* < 0.05 represents a significant difference.

**Figure 6 cells-12-01451-f006:**
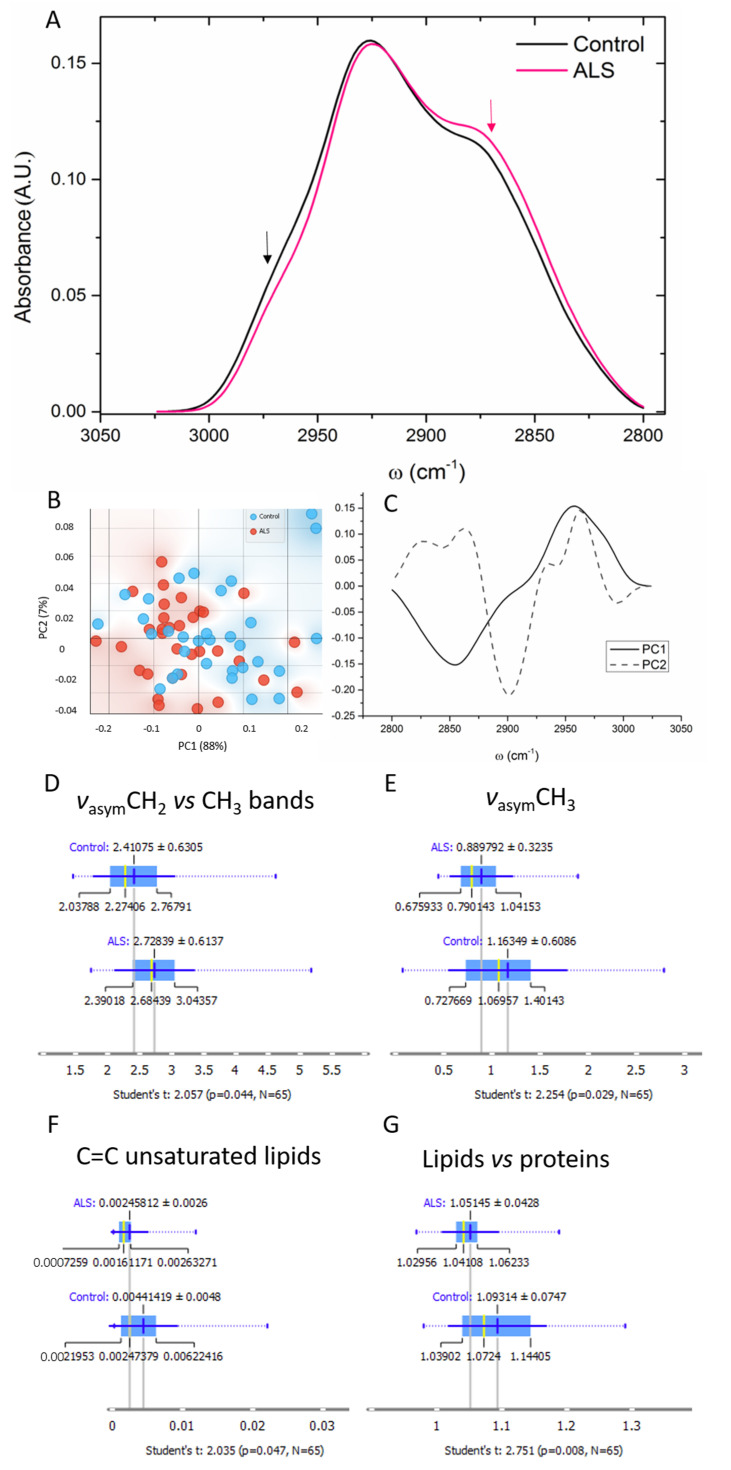
The averaged FTIR spectra of the lipid region 2800–3050 cm^−1^ of ALS (red) and control (black) CSF samples (N = 33, 32, respectively). Bands at ~2870 (red arrow) and ~2960 cm^−1^ (black arrow) are assigned to *ν*_s_CH_3_ and *ν*_as_CH_3_, respectively (**A**). Scatter plot represents the PCA analysis of the first and second principal components (**B**) and the contribution of individual absorbance to the PCAs (loading values) of the first two principal components, PC1 and PC2 (**C**). Ratio calculated of the integrated area of asymmetric CH_2_ vs. CH_3_ bands (**D**), asymmetric CH_3_ bands (**E**), unsaturated C=C bands (3000–3050 cm^−1^) (**F**), and the ratio calculated of the integrated area of total lipids and proteins (**G**).

## Data Availability

The data presented in this study are available on request from the corresponding author.

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
