# Peer review of "Synchrotron-Based Fourier-Transform Infrared Micro-Spectroscopy of Cerebrospinal Fluid from Amyotrophic Lateral Sclerosis Patients Reveals a Unique Biomolecular Profile"

_cells, 2023, doi:10.3390/cells12111451_

Round 1

Reviewer 1 Report

In this work synchrotron-based FTIR spectroscopy and multivariate analysis were explored to analyze cerebrospinal fluid (CSF) from patients with ALS. The authors reported significantly altered IR bands corresponding to lipids, proteins, RNA and DNA, in the FTIR spectra of ALS CSF compared to controls. The paper suggests the applicability of the applied approach in the diagnosis of ALS.

In my opinion, the manuscript would benefit from some revision. Some examples for corrections of Discussion are given below.

- The sentence “Thus, the infrared spectral signature of the RNA is important to evaluate, as the shift in the RNA maximum absorption reflects the structural, conformational change of each biomolecule.” (rows 389-391) should be moved in the beginning of the section “Nucleic acids and carbohydrates” after row 376.

- The phrase “thus suggesting a broader role of these findings in different neurodegenerative diseases.” (rows 397-398) doesn’t sound convincing.

- It is not obvious why FTIR allows detection of changes in a broader group of patients, as well as why this implies more general role in the studied pathology, as stated: “Putatively, the high sensitivity of FTIR employed in our study allows for the detection of glutamate changes in a broader patient group implying a more general role in ALS pathophysiology.” (rows 421-423)

Additionally, I have the following methodological concern:

- It is interesting how the control group was selected. Аs seen in the Table summarizing the clinical data (Supporting material) most individuals included in the control group have serious health problems too, even subjects with MS are included. It is questionable whether this is a good control or not.

- It would be better to use biomolecular instead of bio-macromolecular as throughout the manuscript.

Reviewer 2 Report

Dear authors

The MS entitled “Synchrotron-based Fourier-transform infrared micro-spectroscopy of cerebrospinal fluid from Amyotrophic lateral sclerosis patients reveals a unique macromolecular profile” was thoroughly evaluated. The MS is well written and the most complex spectra has been well resolved. Technically, a difficult task has been undertaken however, the author summed a great deal of work. FTIR for such complex mixtures are very difficult to interpret.

My suggestions and quires are:

In Abstract, capitalize “Fourier transform infrared spectroscopy” as first used.

Q1: How the authors distinguished the OOP bends of C=C in region 800 and 1000 cm-1 from that of RNA/DNA?

Q2: Is the 3D confirmational structure of RNA/DNA functionally significant in ALS?

Q3: What is the significant role of carbohydrate metabolism in ALS?

Q4: Can we assume that the decreased glucose metabolism in as compared to control could be sign of ALS?

Q5: Why the authors not studied the FTIR emerging from Acetylcholine which is the main excitatory neurotransmitter of many brain functions.

References. Check the format of references. It should be according to MDPI format.

Round 2

Reviewer 1 Report

I have no more comments on the revised MS

Reviewer 2 Report

Dear Authors. I am satisfied from the responces.